# Interpreting and Steering LLM Representations with Mutual Information-based Explanations on Sparse Autoencoders

## Abstract

Large language models (LLMs) excel at addressing general human queries, yet they can falter or produce unexpected responses in specific scenarios. Gaining insight into the internal states of LLMs is key to understanding their successes and failures, as well as to refining their capabilities. Recent efforts have applied sparse autoencoders to learn a feature basis for explaining LLM hidden spaces. However, current post-hoc explanation methods can not effectively describe the semantic meaning of the learned features, and it is difficult to steer LLM behaviors by manipulating these features. Our analysis reveals that existing explanation methods suffer from the frequency bias issue, i.e., they tend to focus on trivial linguistic patterns rather than semantics. To overcome this, we propose explaining the learned features from a fixed vocabulary set to mitigate the frequency bias, and designing a novel explanation objective based on the mutual information theory to better express the meaning of the features. We further suggest two strategies to steer LLM representations by modifying sparse feature activations in response to user queries during runtime. Empirical results demonstrate that our method generates more discourse-level explanations than the baselines, and can effectively steer LLM behaviors to defend against jailbreak attacks in the wild. These findings highlight the value of explanations for steering LLM representations in downstream applications.[1]

## 1 Introduction

Large language models (LLMs) have demonstrated strong capabilities in responding to general human requests (Achiam et al., 2023; Dubey et al., 2024; Jiang et al., 2024). Meanwhile, we still often observe failed or unexpected responses in certain situations (Ji et al., 2023; Wei et al., 2024). Gaining insight into the factors behind their successes and failures is crucial for further improving these models. A straightforward way to understand LLM behaviors is directly studying their hidden activations or internal weights. However, it is non-trivial to interpret the hidden states of modern LLMs because of their *polysemantic* nature (Arora et al., 2018; Scherlis et al., 2022), where each dimension of the spaces encodes multiple pieces of unique features. This property allows LLMs to encode more features than the dimensions of their hidden space, but it presents significant challenges for human interpretation and understanding.

Researchers have made significant efforts to overcome the polysemantic challenge. Linear probing (Campbell et al., 2023; Burns et al.; Marks & Tegmark, 2023; Gurnee et al., 2023) is a conventional technique to detect whether an LLM learns a particular feature of interest. Unfortunately, the feasibility of this technique is bounded by its requirement of an annotated dataset with samples including or excluding certain features.

---

[1]We will release our code and data once accepted.

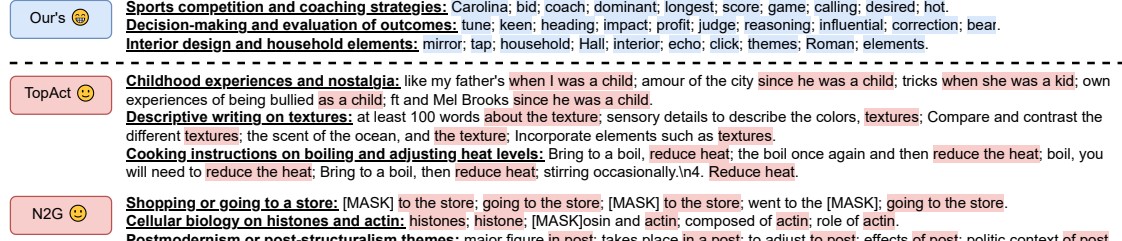

Figure 1: Examples of explanations for a sparse autoencoder trained on Mistral-7b-Instruct. We separate raw extracted spans/words with ";" and boldface the automated summaries. Unlike other methods, our approach tends to produce discourse-level explanations rather than those dominated by rigid linguistic patterns.

To reduce the need for annotated datasets, researchers (Cunningham et al., 2023; Wu et al., 2024; Freire et al., 2024; Bricken et al., 2023) are switching to decomposing the hidden spaces of LLMs in an unsupervised way. In this context, recent research has explored the sparse autoencoder (Olshausen & Field, 1997; Makhzani & Frey, 2013) technique, demonstrating their effectiveness in learning a number of sparse features as a basis to reconstruct the hidden spaces of advanced LLMs with hundreds of billions of parameters from Anthropic (Templeton et al., 2024), OpenAI (Gao et al., 2024), and Google (Lieberum et al., 2024). These *sparse* features are expected to be interpretable, since each feature should only react to a specific kind of content, showing a *monosemantic* nature instead of a polysemantic one.

However, researchers find that the learned sparse features have not shown strong enough explainability to meet our expectations, i.e., understanding LLM encoded features and even steering LLM behaviors. Specifically, Makelov et al. (2024) and Chaudhary & Geiger (2024) designed dedicated tasks to test whether sparse autoencoders could detect sufficient features for certain tasks. However, they found that sparse autoencoders cannot capture enough relevant features to meet these goals, even for simple and experimental-level tasks with clear training samples. Meanwhile, researchers (Gao et al., 2024) also observed that many learned sparse features from advanced LLMs could not be effectively explained with current techniques. These headwinds undermine confidence in extending such techniques to real-world applications.

In this work, we enhance the interpretability and usability of sparse autoencoder features by introducing a new post-hoc explanation method and strategies to steer LLM representations with these features. We first formalize the text generation process with the topic model (Blei & Lafferty, 2006; Arora et al., 2016), revealing that sparse autoencoders learn both *discourse topics* and *linguistic patterns* as features simultaneously, with linguistic patterns being less semantically critical but often dominating. To address this issue, we propose to leverage a fixed vocabulary set to collect explanations and ensure that critical information on learned features is captured based on a mutual information-based objective. We also explore steering LLM representations by modifying the activation of explained features during runtime. Figure 1 shows some examples of explanations generated by our method compared to other explainers, and Figure 2 visualizes our pipeline to steer LLMs with explained features. Experiments on open-source LLMs show that our method provides more meaningful discourse-level explanations, and they are practically usable for downstream tasks. We summarize our contributions as follows:

- Our theoretical analysis identifies a key challenge in explaining learned features from sparse autoencoders, i.e., the frequency bias between the discourse and linguistic features.

- We propose leveraging a fixed vocabulary set to mitigate the frequency bias for explaining learned features. Experimental results show that our method provides more discourse-level explanations than the others.

- We propose steering LLM representations by modifying their activations in response to user inputs during runtime. We apply this approach with our explanations to prevent real-world jailbreak attacks, and show that the steered LLM achieves a significant safety improvement while baseline explanations fail.

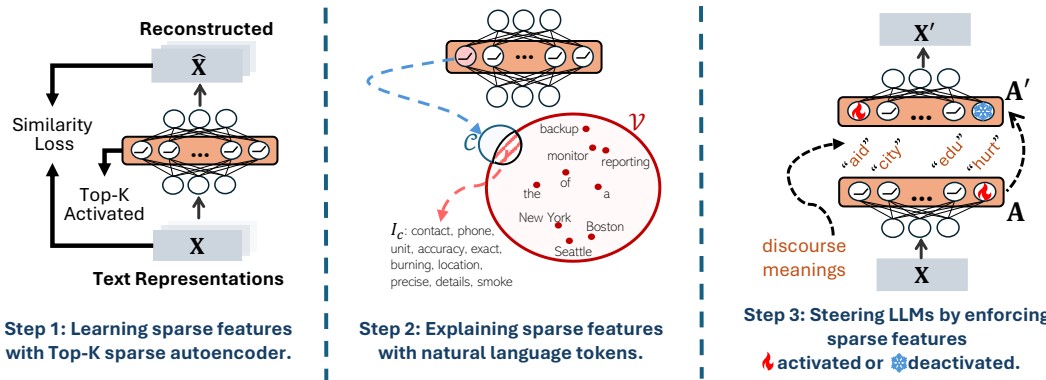

Figure 2: Steering LLM representations with explanations from sparse autoencoders.

## 2 PRELIMINARY

### 2.1 PROBLEM STATEMENT

Let $\mathcal{V}$ denote the vocabulary set, and $X$ be a text of length $N$, where each token $x_n \in \mathcal{V}$ is the $n$-th token of $X$. Given a large language model $f$, the embedding of $X$ at the $l$-th layer is denoted as $\mathbf{X}^{(l)} \in \mathbb{R}^{N \times D}$, where $D$ is latent dimension. In the rest of this paper, we omit superscript $^{(l)}$ for simplification of notations. Our goal is to interpret these embeddings by extracting semantic features from the latent space. We assume that there are $C$ learned feature vectors $\mathbf{W} \in \mathbb{R}^{C \times D}$, so that $\mathbf{X}$ can be decomposed as a linear combination of these feature vectors, i.e., $\mathbf{X} \approx \mathbf{A}\mathbf{W}$, where $\mathbf{A} \in \mathbb{R}^{L \times C}$ are weights of the linear combination for the given instance $\mathbf{X}$. Let $\mathbf{W}_c$ denote the $c$-th row of $\mathbf{W}$. After the decomposition, $\mathbf{X}$ is explainable if we could understand the semantic meaning of each learned feature vector $\mathbf{W}_c$. To achieve this, we aim at seeking a set of words $\mathcal{I}_c \subset \mathcal{V}$ to explain each learned feature $\mathbf{W}_c$ with natural language.

### 2.2 LEARNING AND INTERPRETING LLMS WITH SPARSE AUTOENCODERS

Sparse autoencoders have shown great promise to learn the feature vectors for latent representation decomposition and explaining LLMs in practice (Gao et al., 2024; Lieberum et al., 2024). A standard sparse autoencoder (Olshausen & Field, 1997) is a two-layer multi-layer perceptron $\hat{\mathbf{X}} = \sigma(\mathbf{X}\mathbf{W}) \cdot \mathbf{W}'^{\top}$, where $\mathbf{W}, \mathbf{W}' \in \mathbb{R}^{D \times C}$ are trainable parameters and $\sigma$ refers to the ReLU activation function. Typically, a tight weight strategy is applied, i.e., $\mathbf{W}' = \mathbf{W}$, and the trained weights $\mathbf{W}$ are considered as the learned feature vectors. The traditional training objective of sparse autoencoders can be written as $||\mathbf{X} - \hat{\mathbf{X}}||_2 + \lambda||\mathbf{A}||_1$, where $\mathbf{A} = \sigma(\mathbf{X}\mathbf{W})$ and $\lambda \in \mathbb{R}^+$ is a hyper-parameter to balance the impact of the sparsity constraint. The Top-K sparse autoencoder (Makhzani & Frey, 2013) replaces the ReLU function with the Top-K activation, enforcing each reconstruction to apply with no more than $K$ learned features. Recent studies (Templeton et al., 2024; Gao et al., 2024; Lieberum et al., 2024) have shown that Top-K sparse autoencoders can be used to learn *sparse* features for reconstructing token-level representations from LLMs, where these sparse features are expected to be interpretable by humans.

However, there are limited explorations on collecting a natural language explanation $\mathcal{I}_c$ for each of the learned feature vectors $\mathbf{W}_c$. The most intuitive strategy (Bricken et al., 2023) is collecting some N-gram spans that could best activate the feature vector $\mathbf{W}_c$ over a large corpus. Some researchers (Gao et al., 2024) leverage the Neuron-to-Graph (N2G) algorithm (Foote et al., 2023) to refine the N-gram spans for more precise interpretations. However, it has been found (Gao et al., 2024) that these methods still fail to generate explanations for a large number of learned features from sparse autoencoders trained for LLMs.

## 3 METHODOLOGY

This section first theoretically studies the properties of text generation for learning sparse autoencoders, comparing them to traditional image generation scenarios. With these insights, we propose a mutual information-based post-hoc method to explain the semantics of feature vectors learned by a trained sparse autoencoder. Finally, we design two strategies to steer LLM representations with the explained features.

### 3.1 LEARNING SPARSE FEATURES FROM TEXTUAL DATA

Conventional sparse autoencoders (Olshausen & Field, 1997) are developed based on an assumption for image data, where each image is a linear combination of *features*. A sparse autoencoder learns an *overcomplete* set of visual features, so that any image can be decomposed and reconstructed with the learned features. Early works (Bricken et al., 2023; Cunningham et al., 2023) borrow this framework from image data to textual data, assuming that each token is linearly related to a set of features. However, they ignore some natures of textual data, leading to a suboptimal solution to learning sparse features (Gao et al., 2024).

To start with our theoretical analysis, we consider the text generation task as a dynamic process under the topic-model assumption (Steyvers & Griffiths, 2007; Arora et al., 2016; 2018), where each word $x_n$ is generated at the $n$-th step. This topic model describes a dynamic process in which a person first comes up with a topic $c_n$ they want to express in mind and then selects a word $x_n$ that best represents the topic to say. It means that, in topic models, text generation begins with a predetermined "mind" or theme, guiding word selection to align with that central idea. Similarly, autoregressive language models (Radford et al., 2019) generate text by sequentially predicting each word based on prior context alone, achieving coherence through accumulated context rather than a predefined topic, thus allowing the theme to emerge organically without explicit guidance. Formally, this dynamic process can be driven by the random walk of a discourse vector $\mathbf{e}_{c_n} \in \mathbb{R}^d$ representing what it talks about. The discourse vector $\mathbf{e}_{c_n}$ does a slow random walk at each step $n$, i.e., $\mathbf{e}_{c_n} = \mathbf{e}_{c_{n-1}} + \mathbf{e}_{\epsilon_n}$, where $\mathbf{e}_{\epsilon_n} \sim \mathcal{N}^d(0, \sigma)$. Also, at each step, a word $x_n \in \mathcal{V}$ is sampled based on the discourse vector $\mathbf{e}_{c_n}$. To this end, the text generation process for a sequence of words $X$ is given by:

$$p(X) = \prod_{n=1}^{|X|} p(x_n|c_n) \cdot p(c_n|c_{n-1}). \tag{1}$$

Here, the word emission probability is modelled by $p(x_n|c_n) = \frac{\exp(\langle \mathbf{e}_{x_n}, \mathbf{e}_{\mathbf{c_n}} \rangle)}{\sum_{v \in \mathcal{V}} \exp(\langle \mathbf{e}_v, \mathbf{e}_{c_n} \rangle)}$ (Steyvers & Griffiths, 2007), where $\langle \cdot, \cdot \rangle$ indicates the dot product of two vectors. Since $c_n$ is a random walk of $c_{n-1}$, the topic transmission probability can be computed as $p(c_n|c_{n-1}) = \frac{1}{\sqrt{2\pi} \cdot \sigma} \cdot \exp(\frac{-||\mathbf{e}_{c_n} - \mathbf{e}_{c_{n-1}}||_2}{2\sigma})$ (Olshausen & Field, 1997). Recall that $\mathbf{e}_{c_n} = \mathbf{e}_{c_{n-1}} + \mathbf{e}_{\epsilon_n}$, after a few straightforward derivations, we have

$$\log p(X) \propto \sum_{n=1}^{N} \langle \mathbf{e}_{x_n}, \mathbf{e}_{c_0} \rangle + \sum_{n=1}^{N} \sum_{i=1}^{n} \langle \mathbf{e}_{x_n}, \mathbf{e}_{\epsilon_i} \rangle - \frac{1}{2\sigma} \sum ||\mathbf{e}_{\epsilon_n}||_2. \tag{2}$$

Equation 2 reveals some critical characteristics of textual data that is different from image data. Firstly, there is a shared discourse topic $c_0$ across all words $x_n$ from the same sentence $X$, for $n = 1, ..., N$. However, recent approaches that use sparse autoencoders for LLMs often treat the reconstruction loss for each token independently, without adding constraints to capture the shared concepts. As a result, they fail to isolate the features learned for discourse semantical topics (i.e., $\mathbf{e}_{c_0}$) and linguistic patterns (i.e., $\mathbf{e}_{\epsilon_n}$). In other words, each learned sparse feature may store both discourse and linguistic information, where the latter is less useful for steering LLMs than the previous one. Additionally, discourse topics are rarer than linguistic patterns, as each instance has $N$ times more linguistic patterns than discourse topics, we call it the *frequency bias*. This issue leads to the sparse features that prioritize capturing the linguistic patterns, raising the challenge of interpreting the discourse topics encoded within LLMs.

## 3.2 EXPLAINING LEARNED FEATURES WITH NATURAL LANGUAGE

To interpret the learned features $\{\mathbf{w}_c\}_{c=1}^C$, existing works (Bricken et al., 2023; Gao et al., 2024) typically enumerate a large number of texts, and then treat those whose hidden representations could most activate the learned features as the interpretations. This method works well for interpreting the learned linguistic patterns as they are frequently presented in the corpus, while it is hard to discover the learned discourse topics because the more frequent linguistic patterns dominate (see discussions in Sec. 4.2.2), leading to fail of explaining the amount of learned features (Gao et al., 2024). Since our goal is to understand and control LLM behaviors, we aim to interpret those discourse topics within a feasible budget cost.

To tackle the challenge of frequency bias, we propose to leverage a fixed vocabulary set $\mathcal{V}$ of a general corpus instead of its raw texts. Specifically, our goal is to seek a $K$-word set $\mathcal{I}_c \subset \mathcal{V}$ that can describe most information of the $c$-th feature vector $\mathbf{W}_c$. Mathematically, we let $\mathcal{C}$ denote the knowledge encoded by $\mathbf{W}_c$ and measure the information of $\mathcal{C}$ described by a given word set $\mathcal{V}' \subset \mathcal{V}$ based on their mutual information (Cover, 1999). To this end, the objective of constructing $\mathcal{I}_c$ is defined as

$$
\begin{aligned}
\mathcal{I}_c &= \arg\max_{\mathcal{V}'\subset\mathcal{V},|\mathcal{V}|=K} MI(\mathcal{V}';\mathcal{C}) \propto \arg\min_{\mathcal{V}'\subset\mathcal{V},|\mathcal{V}'|=K} H(\mathcal{C}|\mathcal{V}') \\
&= \arg\max_{\mathcal{V}'\subset\mathcal{V},|\mathcal{V}'|=K} \sum_{\mathbf{c}\in U(\mathcal{C})}\sum_{w\in\mathcal{V}'} p(\mathbf{c})p(w|\mathbf{c})\log p(\mathbf{c}|w),
\end{aligned}
\tag{3}
$$

where $U(\mathcal{C})$ are all possible vectors that express the knowledge $\mathcal{C}$. Since we obtain $\mathbf{W}_c$ by training a sparse autoencoder—and ideally, each learned feature vector encodes a unique piece of knowledge—we assume that $p(\mathbf{c} = \mathbf{W}_c) \approx 1$ and $p(\mathbf{c} \neq \mathbf{W}_c) \approx 0$. This allows us to simplify the expression:

$$
\mathcal{I}_c^* \propto \operatorname{argmax}_{\mathcal{V}'\subset\mathcal{V},|\mathcal{V}'|=K} \sum_{w\in\mathcal{V}'} p(w|\mathbf{W}_c)\log p(\mathbf{W}_c|w).
\tag{4}
$$

By leveraging word embedding $\mathbf{e}_w$ of word $w$, we empirically estimate $p(w|\mathbf{W}_c)$ and $p(\mathbf{W}_c|w)$ by

$$
p(w|\mathbf{W}_c) = \frac{exp(\langle\mathbf{e}_w,\mathbf{W}_c\rangle)}{\sum_{w'\in\mathcal{V}} exp(\langle\mathbf{e}_{w'},\mathbf{W}_c\rangle)}, \quad p(\mathbf{W}_c|w) = \frac{exp(\langle\mathbf{e}_w,\mathbf{W}_c\rangle)}{\sum_{c'\in\mathcal{C}} exp(\langle\mathbf{e}_w,\mathbf{W}_{c'}\rangle)}.
\tag{5}
$$

Compared with a trivial strategy that simply obtains $K$ words whose embeddings maximally activate the feature vector, this mutual information-based method reveals the importance of normalizing activations of a single word across all learned features. In other words, if a word embedding constantly leads to a significant large dot product with all features, the word will not express enough specificity to any certain feature. TF-IDF (Salton & Buckley, 1988) is a practical technique for mitigating frequency bias. As noted by Aizawa (2003), TF-IDF can be formulated from the same mutual information-based objective that we used in this work. However, it relies on assumptions about word distributions over documents, which do not hold in our feature interpretation task. Thus, our method is derived from a more general perspective, better aligning this objective with interpreting learned sparse feature vectors.

## 3.3 STEERING LLMs WITH EXPLAINED FEATURES

Given learned features $\{\mathbf{w}_c\}_{c=1}^C$ and their explanations $\{\mathcal{I}_c\}_{c=1}^C$, we could identify a subset of the features $\mathcal{S} = \{\mathbf{w}_s\}_{s=1}^S \subset \{\mathbf{w}_c\}_{c=1}^C$ that are correlated with a specific LLM behavior we are interested in based on their explanations (e.g., harmful knowledge or safety awareness in our study). This process can be either manually or automatically (Bills et al., 2023). Considering the hidden representations of an input prompt as $\mathbf{X}$, we propose two strategies to steer LLM representations with the identified features $\mathcal{S}$ during runtime.

**Amplification.** We amplify $\alpha$ times of the activations on our identified feature vectors, i.e., $\mathbf{X}' = \mathbf{X} + \alpha \cdot$ ReLU$(\mathbf{XS})\mathbf{S}^\top$, where $\mathbf{S}$ is matrix form of identified set $\mathcal{S}$, and $\alpha$ is a hyper-parameter. We encourage LLMs

to be more aware of the identified features if $\alpha > 0$, and pay less attention to them if $\alpha < 0$. Especially, $\alpha = -1$ indicates that we erase the LLM's awareness of the identified features.

**Calibration.** We enforce LLMs to focus on the identified features to a certain level $\beta$, i.e., $\mathbf{X}' = \mathbf{X} - \text{ReLU}(\mathbf{XS})\mathbf{S}^\top + \beta \cdot \bar{\mathbf{s}}$, where $\bar{\mathbf{s}} = \frac{1}{S} \sum \mathbf{w}_s$ is the mean vector of $\mathcal{S}$ and $\beta$ is a hyper-parameter. This strategy basically shifts the LLM's hidden space toward the center of our target feature vectors.

The above two strategies are responsible for different purposes of steering LLMs, and they could work together. We would also emphasize that the proposed strategies are efficient as we only monitor a subset of our interested features $\mathbf{S}$ instead of the entire set of learned sparse features $\mathbf{W}$.

## 4 EXPERIMENTS

This section investigates two research questions. RQ1: Does the proposed method generate more discourse-level explanations than traditional methods? RQ2: Whether these discourse-level explanations are useful in steering LLM behaviors? To answer these questions, we first train a Top-K sparse autoencoder for open-sourced LLMs as our foundation (Sec. 4.1). We then compare the explanations of the trained sparse autoencoder with our proposed and other explanation methods for RQ1 (Sec. 4.2). We finally explore the usability of these explanations for downstream tasks, i.e., jailbreak defense, for RQ2 (Sec. 4.3).

### 4.1 GENERAL SETTINGS

**Language Models.** In this work, we study LLMs from the Mistral family (Jiang et al., 2023) as it has demonstrated its strong usability in the wild. In particular, we choose the Mistral-7B-Instruct model. We follow the settings from previous work (Lieberum et al., 2024) to select our target layer. In their work, they train SAEs with hidden representations from the 10th, 21st, and 32nd layers of Gemma2-9B-Instruct. Considering Gemma2-9B-Instruct consists of 42 layers, these numbers roughly refer to the first 24%-th, 50%-th, and 76%-th layers, respectively. In addition, since our goal is to steer LLM predictions and researchers (Nostalgebraist, 2020) have observed that LLMs begin performing next-word prediction tasks in their shallow layers, we seek a shallow layer to leave room for changing LLM predictions. To this end, we choose the most shallow layer from (Lieberum et al., 2024), i.e., the 24%-th layer of the entire model, which refers to the 8th layer of Mistral-7B-Instruct with 32 layers in total. Without specifics, the greedy search decoding with a maximum of 512 new tokens is applied to our experiments for reproducibility.

**Datasets.** Since our goal is to develop sparse autoencoders for understanding and controlling LLMs for different applications, we select various instruction-tuning datasets for training our backbone sparse autoencoder. In specific, we contain the training subset of the ShareGPT (RyokoAI, 2023), UltraChat (Ding et al., 2023), HH-RLHF (Bai et al., 2022), WebGLM-QA (Liu et al., 2023), Evol-Instruct (Xu et al., 2023), and HelpSteer2 (Wang et al., 2024) datasets. For the UltraChat dataset, we randomly sample 400K instances from its training subset. We also drop duplicate prompts across different datasets. To this end, we have retained about 711K unique user queries covering diverse topics and user intents. We randomly select 90% of samples to form our training set, and the rest is our validation set. Overall, we collect 113M tokens for training and 12M tokens for validating, with an average length of 177.9 tokens per query.

**Training Details.** Our training procedures and hyper-parameter settings majorly follow the previous works (Bricken et al., 2023; Gao et al., 2024; Lieberum et al., 2024). Specifically, we initialize $C = 2^{16}$ feature vectors for a Top-K sparse autoencoder with Kaiming initialization (He et al., 2015). Here, $C = 2^{16}$ is set according to the scaling law between the number of features $C$ and the number of training tokens $Z$ found by Gao et al. (2024), i.e., $C = \mathcal{O}(Z^\gamma)$, where $\gamma \approx 0.60$ for GPT2-small and $\gamma \approx 0.65$ for GPT-4.[2]. Appendix B provides more detailed settings about training SAEs for our subject LLM.

---

[2] Empirically, $\gamma \approx 0.5978$ in our study.

Table 1: Qualitative analysis on generated explanations. Both TopAct and N2G tend to collect raw explanations sharing the same word-level patterns, while our method captures more discourse-level explanations.

| Method | Automated Summary | Raw Explanation |
|---|---|---|
| **Ours** | Art evaluation and critique. | commonly; impact; cater; widely; normally; gallery; judge; pros; independent; accurately |
| | Analysis of performance metrics. | landscape; graph; retirement; performance; communication; density; cut; golf; measure; measures |
| | Temporal concepts and sequences in narratives. | previously; suddenly; repeated; history; once; initially; nearest; already; normally; originally |
| **TopAct** | Evaluation criteria for assessments or analyses in various contexts. | What criteria does Pitchfork use to; What evaluation criteria will Kumar organization use to; and what criteria were used; market? what specific criteria should be used; needed to conduct a comprehensive analysis and the criteria used |
| | Instructional prompts or commands for providing steps in a process. | [INST] Provide step; [INST] Provide step; [INST] Provide step; [INST] Provide step; [INST] Provide step |
| | Repetition of the word "again" in various contexts | ideas and produce compelling content — again; Pine View School again; technologies segment is again; pushed on the ceiling,and again; Echoed through the valley, again |
| **N2G** | Data format: Comma-Separated Values (CSV). | CSV; CSV; CSV; CSV; csv[MASK] |
| | Scheduling and managing appointments. | schedule appoint; upcoming appoint[MASK]; appointment; appointment; upcoming appoint[MASK] |
| | Video game titles. | Final Fant; Final Fant; Final Fant; Final Fant; Metal Gear |

**Explanation Baselines.**    Our study considers several existing works for sparse autoencoder explanations as baselines. *TopAct* (Bricken et al., 2023) collects a mount of text spans from the corpus that could maximally activate it. *N2G* (Gao et al., 2024) steps further by masking some words from the activated spans that show limited contributions to the activations. We collect their activated spans, with a maximum of 10 tokens, over the entire validation set, and we keep the most activated span from each entry to increase their diversity.

### 4.2  EVALUATING EXPLANATIONS OF SPARSE FEATURES

Exactly measuring the explanation quality of features from sparse autoencoders is still an open question (Rajamanoharan et al., 2024b). One that is commonly applied is conducting human studies (Bricken et al., 2023; Rajamanoharan et al., 2024a; Gao et al., 2024; Rajamanoharan et al., 2024b), where the human subjects are asked to determine whether an explanation is meaningful or not. We follow this paradigm to evaluate the explanations from different methods, and we scale up this process by replacing human subjects with GPT-4o as existing works (Bricken et al., 2023; Bills et al., 2023; Rajamanoharan et al., 2024b).

### 4.2.1  EXPERIMENTAL DESIGNS

We conduct both qualitative and quantitative analyses of the explanations with the help of our machine annotator. Given a feature vector and its raw explanations, the machine annotator is called to provide a short summary of the explanations with an option to say "Cannot Tell" in case the raw explanations make no sense (please check details in Appendix. A). Here, the raw explanations of TopAct and N2G are the top-5 most activated text spans, while our method chooses the top-10 words over a vocabulary set consisting of the 5000 most common words in the training set. It is crucial to recognize that the vocabulary set used for collecting

explanations does not have to be the built-in vocabulary sets of LLMs. Specifically, we tokenize the words from the raw training data with regular expressions and only keep those words with English alphabets, digital numbers, and simple connection symbols. Once the summary is collected, we call the machine annotator in a new thread to judge whether the raw explanations are relevant to the given summary. We follow previous work (Rajamanoharan et al., 2024b) to give the judgment with some options, namely "yes", "probably", "maybe", and "no", where in our study, we treat the summaries are judged with "yes" or "probably" as successfully explained. Table 1 shows some randomly selected cases with a judgment "yes" and the text spans or words are separated with the symbol ";" (please check more cases in Appendix C). We also report the percentage of successfully explained the raw explanations from various explainers in Table 2.

### 4.2.2 Results

**TopAct and N2G tend to collect text spans sharing the same lexical patterns, while our method prefers words sharing a concise topic.** In Table 1, we could first see that these explanations marked with "yes" are highly interpretable, demonstrating the effectiveness of using machine annotators to replace human annotators for scaling up the evaluation process. While both baselines and our proposed method generate reasonable explanations, we also find some different characters from their raw explanations. In specific, the raw explanations of TopAct or N2G typically share the same linguistic phrases, such as "used to" for the first case of TopAct and "CSV" for the first case of N2G. However, the selected words with our method do not appear as such lexical-level phrases; instead, the group of them illustrates a concise topic. This difference highlights the motivation of our research to find discourse-level explanations.

**Our method generates more reasonable explanations than that of TopAct and N2G.** Table 2 reports the percentage of learned sparse features that are successfully explained, and we group them by those that have been activated from the validation set or overall. We observe that many learned features haven't been reasonably explained with TopAct or N2G because not enough patterns have been activated on the validation set, which is one of the drawbacks of relying on activating input text for generations. One may argue that we can collect activated spans from the training set. However, these activated patterns can be significantly biased, as the sparse autoencoder is supposed to overfit the training set (Tom & Chris, 2023). Preparing a large validation set to ensure each learned sparse feature collects enough activation spans weakens the usability of

Table 2: Explanation rates of learned sparse features on the features only activated validation set or overall features.

| Method | Explanation Rate | |
| --- | --- | --- |
| | Activated | Overall |
| **TopAct** | 59.16 | 23.17 |
| **N2G** | 38.79 | 15.13 |
| **Ours** | 67.39 | 66.98 |

these methods again. Even only considering the learned features that have been activated on the validation set, the proposed method shows a stronger explainable rate than the baselines. It is not surprising that N2G actually provides worse raw generations than TopAct, as we found evidence[3] that N2G shows a stronger preference for lexical patterns than TopAct, even if they are fake ones. These observations showcase the challenge of interpreting the discourse-level meanings behind the learned sparse features.

### 4.3 Using Explained Features for Downstream Tasks

This section considers jailbreak defense as a downstream application to utilize our explained features. Our goal is to defend jailbreak attacks while keeping its helpfulness in responding to normal queries. We choose this task because of its generalizability across different scenarios that need to deploy LLMs. Also, existing defense strategies haven't shown practical utility due to their poor effectiveness or unbearable latency.

---

[3]For example, one sparse feature whose raw explanation of TopAct is "6th century (via History Magazine). Before that"; "Prior to Chomsky's work,"; and "Reference [2]: Before the GPS,".It is clear that this feature captures "referring related works". However, N2G simplifies them to "Before that"; "Prior to [MASK]omsky's work"; and "Before [MASK] GPS,", which obviously changes the meaning and concentrates on some trivial patterns, i.e., "Before" and "Prior to".

Table 3: Defending Mistral-7b-Instruct from jailbreak attacks without model training. The Salad-Bench reports the attack success rate (ASR) to illustrate the effectiveness of different models to prevent jailbreak attacks, while the MT-Bench shows its automatic scoring results on the helpfulness of general user queries.

| Category | Method | Salad-Bench (Safety) | | MT-Bench (Helpful) | |
| | | ASR (↓) | Time (↓) | Score (↑) | Time (↓) |
|---|---|---|---|---|---|
| w/o Defense | | 81.6 | 1.0x | 6.5 | 1.0x |
| **Perturbation** | Random Patch | 80.6 | 4.9x | 3.8 | 1.6x |
| | Random Insert | 79.4 | 6.5x | 3.7 | 1.6x |
| | Random Swap | 73.8 | 5.6x | 3.0 | 1.6x |
| | Self-Robustness | 16.2 | 6.9x | 5.3 | **16.9x** |
| **Prompting** | SafePrompt | 79.0 | 1.0x | 6.5 | 1.0x |
| | XSafePrompt | 77.8 | 0.9x | 6.1 | 0.9x |
| | Self-Reminder | 73.0 | 0.9x | 6.3 | 0.9x |
| **SAE Steer** (Ours) | Erase Harmful (EH) | 81.0 | 1.0x | 5.9 | 1.0x |
| | Aware Security (AS) | 73.2 | 0.8x | 6.0 | 0.9x |
| | EH + AS | 72.8 | 0.8x | 5.9 | 0.9x |

### 4.3.1 EXPERIMENTAL DESIGNS

We leverage two benchmarks to evaluate our downstream task performance. In specific, Salad-Bench (Li et al., 2024) is introduced to evaluate the safety of LLMs, and MT-Bench (Zheng et al., 2023) is applied to evaluate their general helpfulness. Two categories of the defense strategies that do *not* require any training datasets are considered as the baseline methods, where the *perturbation-based* methods include Random Patch/Insert/Swap (Robey et al., 2023) and Self-Paraphrase (Cao et al., 2023), and the *prompting-based* methods include SafePrompt/XSafePrompt (Deng et al., 2023), and Self-Reminder (Xie et al., 2023). Since most of the perturbation-based baselines are time-consuming, we randomly select 10% of the samples to conduct a smaller test set for all our evaluations. Note that all baselines and our methods will not be trained on any data in this experiment. The attack success rate (ASR) on Salad-Bench, GPT-4o-mini evaluated MT-Bench scores, and the normalized consuming time are listed in Table 3.

We can consider three specific strategies for jailbreak defense with the proposed Amplification and Calibration methods. (1) Erase Harmful (EH) monitors whether any "*harmful*" features are activated, and *erase* them if so. (2) Aware Security (AS) consistently activates those *safety* features during responding. (3) Applying both AS and EH strategies at the same time. Here, we follow the hazard taxonomy of Llama3-Guard (Llama Team, 2024) to judge whether each feature is harmful. Inspired by this hazard taxonomy, we manually craft a safeguarding taxonomy listing 7 categories to classify safety strategies. We prompt the machine annotator to provide the harmfulness and safety labels for each learned feature by providing their explanations. To ensure quality, we only consider the learned features with the explainable label "yes". As a result, our method selects 141 and 48 features for the AS and EH strategies, respectively. For hyper-parameter $\beta$ of AS, we grid search some numbers and find that 2.5 shows the best practice in balancing safety and overall helpfulness. Table 3 and Figure 3 report the results with our and baseline explanations, respectively. Appendix D provides a case study on defending jailbreak attacks with the AS strategy.

### 4.3.2 RESULTS

**Sparse autoencoder can steer LLMs behavior during runtime.** First of all, we can read from Table 3 that all perturbation-based defense strategies are not practical for real-world use, as they either significantly compromise overall helpfulness or introduce intolerable latency. In contrast, most prompting-based methods maintain general helpfulness but fail to defend against jailbreak attacks. The notable exception is the state-of-the-art baseline, Self-Reminder, which achieves safety and helpfulness within the same computing budget.

Compared with them, our proposed sparse-autoencoder-based methods exhibit a strong jailbreak defense ability (Salad-Bench: $81.6 \rightarrow 72.8$) with only a minor reduction on helpfulness (MT-Bench: $6.5 \rightarrow 6.0$). The success of our method in such a challenging task provides a promising direction for other scenarios.

**The key to preventing jailbreak attacks is not to forget harmful knowledge, but to enhance safety awareness.** One interesting finding from our experiment is that the strategy of erasing harmful knowledge has no significant contribution to the jailbreak defense, contradicting our intuitive understanding of the jailbreak defense. The significant improvement of our Aware Security strategy for jailbreak defense actually aligns with the main idea of Self-Reminder – "remind ChatGPT to respond responsibly" (Xie et al., 2023).

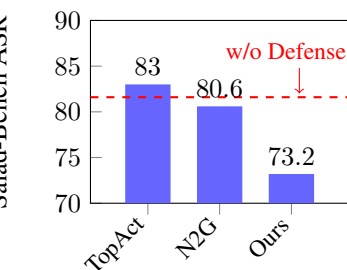

We also apply the Aware Security strategy to the TopAct and N2G explanations and report their results in Figure 3. Only N2G shows a slight reduction in ASR versus the no-defense baseline. We have tuned $\beta$ but cannot see a clear improvement. One possible reason is that their selected safety strategies are too lexical-level and fine-grained. Here is an example feature that has been annotated with a "Physical Defense Category" as its summary is "Locking mechanisms or security systems" with a raw explanation: "locks; locks; lock; have a two-stage lock; lock." To compare with, one of our method annotated with the same category has a summary of "Emergency response and location tracking" with a raw explanation "contact, phone, unit, accuracy, exact, burning, location, precise, details, smoke." These observations highlight our motivation to explain discourse-level features.

Figure 3: Applying Aware Security for jailbreak defense based on explanations from different methods.

## 5 RELATED WORKS

Modern large language models have shown promising text-generation abilities, prompting researchers to explore their internal mechanisms. One approach (Belinkov et al., 2018; Jawahar et al., 2019; Rogers et al., 2021) develops contrastive datasets to probe hidden states for specific features, but it is limited by the polysemantic nature of neurons (Elhage et al., 2022; Olah et al., 2020), making the explanations non-concise and difficult to apply in downstream tasks. To overcome this, researchers (Bricken et al., 2023) propose learning orthogonal basis vectors to better understand LLMs. Early works (Beren & Black, 2022; Wu et al., 2024) applied singular vector analysis to identify concise, interpretable directions in neuron activations. Soon after, sparse autoencoders (Bricken et al., 2023; Cunningham et al., 2023) were introduced, allowing for a more flexible settings. Sparse autoencoders, initially used to analyze image data (Olshausen & Field, 1997; Makhzani & Frey, 2013), are now being applied to LLMs. Researchers from Anthropic (Bricken et al., 2023) and EleutherAI (Cunningham et al., 2023) demonstrated that activations from smaller models like GPT-2 and Pythia yield highly interpretable features. Subsequent studies showed these features help interpret model behaviors in tasks like indirect object identification (Makelov, 2024), translation (Dumas et al.), and circuit detection (Marks et al., 2024). Recent works (Templeton et al., 2024; Gao et al., 2024; Lieberum et al., 2024) confirm this technique's success with larger LLMs. Our study follows this path, and advances by developing a method for generating discourse-level explanations to steer LLM representations.

## 6 CONCLUSIONS

This study steps a solid stamp toward understanding and steering LLM representations in the wild. Our theoretical analysis first reveals a frequency bias between discourse and linguistic features learned by sparse autoencoders. To eliminate this bias, we propose seeking words from a fixed vocabulary set and designing a mutual-information-based objective to ensure the collected words capture the features' meanings. Additionally, we demonstrate that our steering strategies effectively enhance the safety of LLMs using our mutual-information-based explanations, while baseline methods fail to achieve the same. Overall, this study underscores the importance of discourse-level explanations in effectively controlling LLM behavior.

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

## A SCALING UP WITH MACHINE ANNOTATORS

We build on recent progress in automated interpretation (Bills et al., 2023; Chaudhary & Geiger, 2024; Gao et al., 2024; Lieberum et al., 2024) by utilizing advanced large language models to replicate human annotators in producing high-level interpretations. This approach allows us to leverage machine annotators, enabling us to scale our methods for analyzing the entire model and yielding more robust results.

We employ GPT-4o-mini[4] as our machine annotator. Our experiments utilize the gpt-4o-mini-2024-07-18 model with a hyper-parameter temperature=0 for greedy decoding. For each response, we allow a maximum of 1024 tokens. To ensure the quality of automatic annotation, we design our prompting template with both the role-playing strategy and presenting in-context examples. We list our prompting template for our word-list-based explanation summarization and the explainability judgment as follows.

### A.1 TEMPLATE 1

We directly append the words to this template to annotate the summary of the raw explanations with 10 selected words from our proposed method. In this template, we start with placing the role-play instruction in the system prompt. We then provide heuristic examples to simulate a multi-turn conversation between a user and an agent. In this way, once we attach the new word list-based raw explanations from our method to this template, the machine annotator will directly generate the summarization for this explanation.

---

**Template-1 for Automated Summary with Word-based Raw Explanations**

```
System: You are studying a neural network. Each neuron looks for one
particular concept/topic/theme/behavior/pattern. Look at some words
the neuron activates for and guess what the neuron is looking for.
Pay more attention to the words in the front as they supposed to be
more correlated to the neuron behavior. Don't list examples of words
and keep your summary as detail as possible. If you cannot summarize
most of the words, you should say "Cannot Tell."

User: accommodation, racial, ethnic, discrimination, equality, apart,
utterly, legally, separately, holding, implicit, unfair, tone.
Agent: Social justic and discrimination.

User: B., M., e., R., C., OK., A., H., D., S., J., al., p., T., N.,
W., G., a.C., or, St., K., a.m., L..
Agent: Cannot Tell.

User: Scent, smelled, flick, precious, charm, brushed, sealed, smell,
brace, curios, sacred, variation, jewelry, seated.
Agent: Perception of scents and precious objects.

User: BP, HR, RR, O2 Sat, T, Ht, UO, BMI, BSA.
Agent: Medical measurements in emergency rooms.
```

---

[4]https://platform.openai.com/docs/guides/gpt



**Template-1 for Automated Summary with Word-based Raw Explanations (continued)**

```
User: actual, literal, real, Really, optical, Physical, REAL,
virtual, visual.
Agent: Perception of reality.

User: Go, Python, Java, c++, python3, c#, java, Ruby, Swift, PHP.
Agent: Morden programming language.

User: 1939-1945, 1945, 1942, 1939, 1940, 1941.
Agent: Years of the second world war.

User: 1976, 1994, 1923, 2018, 2014, 1876, 1840.
Agent: Cannot Tell.

User:
```



## A.2 TEMPLATE 2

Once we collect the summary of the raw explanation with the previous prompt, we then call the machine annotator again in a separated thread to evaluate whether the summary is hallucinated or not by using the following prompting template, where the placeholders "Summary" and "Raw Explanation" will be filled with real data. Note that if the machine annotator gives "Cannot Tell" as its prediction in the summarization stage, we will directly set the judgment for this task as "No".



**Template-2 for Summary Judge with Word-based Raw Explanations**

```
System: You are a linguistic expert. Analyze whether the words well
represent the concept/topic/theme/pattern. Organize your final
decision in format of "Final Decision: [[Yes/Probably/Maybe/No]]".

User: Concept/Topic/Theme/Pattern: {Summary}.
Words: {Raw Explanation}.
Agent:
```



## A.3 TEMPLATE 3

Since baseline explainers (TopAct and N2G) consider N-gram spans as raw explanations, we found that the previous word-list-based prompting template leads a poor performance for their interpretability. Thus, we followed the strategies before to define the following text-span-based prompting templates. Here, the in-context examples of text spans are collected from previous work (Bricken et al., 2023). Specifically, similar to using Template 1 to summarize our extracted raw explanations, we append the extracted text spans from TopAct or N2G to this template. Note that we numerate each extracted span with a unique index.

---

**Template-3 for Automated Summary with Span-based Raw Explanations**

```
System: You are studying a neural network. Each neuron looks for one
particular concept/topic/theme/behavior/pattern. Look at some spans
the neuron activates for and guess what the neuron is looking for.
Pay more attention to the [last few words] of each spans in the front
as they supposed to be more correlated to the neuron behavior. Ignore
the [MASK] patterns in the spans. Don't list examples of spans and
keep your summary as detail as possible. If you cannot summarize
most of the spans, you should say "Cannot Tell."

User: Span 1: w.youtube.com/watch?v=5qap5aO4z9A
Span 2: youtube.come/yegfnfE7vgDI
Span 3: {'token': 'bjXRewasE36ivPBx
Span 4: /2023/fid?=0gBcWbxPi8uC
Agent: Base64 encoding for web development.

User: Span 1: cross-function[MASK]
Span 2: cross-function
Span 3: [MASK][MASK] cross-function\n
Agent: Particular phrase 'cross-function'.

User: Span 1: novel spectroscopic imaging platform
Span 2: and protein evolutionary network modeling
Span 3: reactions-centric biochemical model
Span 4: chaperone interaction network
Agent: Biological terms.

User: Span 1: is -17a967
Span 2: what is 8b8 - 10ad2
Span 3: 83 -11111011001000001011
Span 4: is -c1290 - -1
Agent: Synthetic math: Arithmetic, numbers with small digits,
in unusual bases.

User:
```

## A.4 TEMPLATE 4

We evaluate the quality of automated summarization using almost the same as Template 2, where we only
change the phrase from "word" to "span" to fit the format of raw explanations from the baseline explainers.

---

**Template-4 for Summary Judge with Span-based Raw Explanations**

```
System: You are a linguistic expert. Analyze whether the text spans
well represent the concept/topic/theme/pattern. Organize your final
decision in format of "Final Decision: [[Yes/Probably/Maybe/No]]".

User: Concept/Topic/Theme/Pattern: {Summary}.
Spans: {Raw Explanation}.
```

## B   TRAINING SPARSE AUTOENCODERS ON MISTRAL-7B

Our training procedures and hyper-parameter settings majorly follow the previous works (Bricken et al., 2023; Gao et al., 2024; Lieberum et al., 2024). Specifically, we initialize $C = 2^{16}$ feature vectors for a Top-K sparse autoencoder with Kaiming initialization (He et al., 2015). Here, $C = 2^{16}$ is set according to the scaling law between the number of features $C$ and the number of training tokens $Z$ found by Gao et al. (2024), i.e., $C = \mathcal{O}(Z^\gamma)$, where $\gamma \approx 0.60$ for GPT2-small and $\gamma \approx 0.65$ for GPT-4.[5] To prevent dead neurons, we also apply the tied-weight strategy as suggested by Gao et al. (2024). We use Adam optimizer (Kingma, 2014) with a constant learning rate of $1e^{-3}$ and epsilon of $6.25e^{-10}$ to train a total of 4 epochs. The hyper-parameters $\beta_1$ and $\beta_2$ of the optimizer are $0.9$ and $0.999$ following Gao et al. (2024), respectively. We set the batch size as 512 queries, leading to around 90K tokens per gradient update, which is the same volume as Gao et al. (2024). The mixed precision training strategy (Micikevicius et al., 2017) is also applied to speed up the training process as Lieberum et al. (2024) found that it only shows a slightly worse impact on the model performance. Top-K sparse autoencoder has an initial sparsity $K = 200$, and it gradually decreases to the target sparsity $K = 20$ in the first 50% training samples of the first epoch. The training process runs on one Nvidia A6000 GPU with CUDA 12.6 and takes about 16 hours per epoch.

## C   QUALITATIVE ANALYSIS ON RAW EXPLANATIONS

This section first provides an extension to our qualitative analysis of the raw explanations generated by different methods discussed in Section 4.2.2. In particular, Table 4, Table 5, and Table 6 provide more raw explanations and their automated summarization from Ours, TopAct, and N2G, respectively.

### C.1   ANALYSIS TO RAW EXPLANATIONS FROM OURS

The extended qualitative analysis on Ours demonstrates the robustness of our method in generating discourse-level explanations. Table 4 showcases a wide variety of explanations that extend beyond mere lexical overlaps, instead providing meaningful insights into different topics or concepts. For instance, explanations such as "Botanical classification and gardening practices" and "Urban development and community engagement" encapsulate coherent themes that align well with their raw explanations, reflecting the interpretative depth of our approach. This contrasts sharply with the baseline methods, which often focus on repetitive patterns or word-level constructs. By leveraging a fixed vocabulary set and mutual information-based objective, our method avoids frequency biases and captures semantically rich discourse features.

### C.2   ANALYSIS TO RAW EXPLANATIONS FROM BASELINES

The extended qualitative analysis of the baselines TopAct and N2G highlights their tendencies to focus on repetitive linguistic patterns and fine-grained lexical constructs rather than capturing broader semantic or discourse-level themes. As shown in Table 5, TopAct often generates explanations dominated by repetitive queries or descriptive patterns, such as "What types of medical facilities are available for" or "Discuss the impact of social media on." While these patterns are interpretable, they largely lack thematic depth, emphasizing lexical regularities over conceptual diversity. On the other hand, in Table 6, N2G explanations successfully identify the most critical parts of the raw explanations and omit those non-critical ones with "[MASK]", resulting in a shortened raw explanations than the TopAct. However, N2G still falls short of representing more complex and discourse-level features. This limitation underscores the advantage of our proposed method in moving beyond the frequency bias to capture more coherent and meaningful features.

---

[5]Empirically, $\gamma \approx 0.5978$ in our study.

Figure 4: A case study on steering LLMs to defense jailbreak attack by Aware Security (AS).

## D  CASE STUDY ON STEERING MODEL BEHAVIORS

We provide a case study in Figure 4 on defending against jailbreak attacks using our proposed method. Specifically, we follow the aware security strategy introduced in Section 4.3.1 to perform the jailbreak defense. The attacking prompt comes from the Salad-Bench (Li et al., 2024) with a role-play attacking strategy, where the attacker asks the LLM to play in an "opposite mode" so that it will be misleading to generate some dangerous advice to the users about using the microwave. In specific, we could observe that the original LLM follows the instructions from the attacker to suggest that the user blow up items in the microwave within the "opposite mode" (e.g., "[AntiGPT]"). There is no doubt that this response is harmful and unsafe to the users, indicating a successful attempt from the attacker.

However, by constantly enforcing the security-aware features to be activated at a level of $\beta = 1$, we observe that the original response becomes less harmful, where the LLM specifies that the blow-up items should be some foods, such as "marshmallows, popcorn, and hot dogs". Finally, when we enforce the activations to a more significant level, i.e., $\beta = 3$, the LLM entirely rejects the harmful premise of the prompt, providing a response that strictly adheres to safety guidelines. Specifically, the LLM refuses to engage with the idea of "blowing up items" in a microwave, emphasizing the importance of following safety protocols and avoiding any unsuitable items. By activating security-related features more strongly, the method demonstrates the capability not only to mitigate harmful responses but also to completely align the model's output with ethical and safety standards. This case study illustrates the effectiveness of our strategy in steering the LLM's behavior towards responsible and safety-conscious outputs.

Table 4: Extended qualitative analysis on generated explanations from our proposed method.

| Method | Automated Summary | Raw Explanation |
|---|---|---|
| Ours | Local business and community engagement. | weekly, regional; native; pros; locally; good; cater; blog; perform; shop |
| | Botanical classification and gardening practices. | flower; hybrid; border; composition; popular; origin; habits; commonly; divide; fit |
| | Influence and alignment of ideas or concepts. | turn; impact; aligned; turning; leading; surrounding; nature; highlight; ideas; align |
| | Diverse strategies and approaches in chatbot development and interaction. | differently; pros; thorough; tricks; observations; view; approaches; Eastern; strategies; chatbot |
| | Digital solutions and services for businesses. | meaningful; inclusive; durable; online; tracking; quick; instant; hosting; marketing; processing |
| | Music education and authentic musical experiences. | stake; genuine; musical; authentic; arrangements; composition; classes; lessons; friend; empower |
| | Processes of change and interaction in systems or relationships. | crack; returning; describe; emerging; transform; transport; mutual; accompanied; interactions; index |
| | Personal development and productivity strategies. | cycle; trial; productive; lessons; lifestyle; neutral; Academy; rhythm; goal; goals |
| | Culinary arts and craftsmanship. | construction; variety; manual; design; fit; dinner; brand; craft; lunch; um |
| | Detection and identification of problems in the context of surveillance or monitoring systems. | detect, detective, detected, early, heat, instant, problem, parking, identifying, detection |
| | Urban development and community engagement. | productivity; interesting; align; correspond; hub; housing; grant; surrounding; mix; inform |
| | Impact of jazz music on youth and critical awareness. | best; question; contributing; mind; jazz; stake; critics; critique; kids; awareness |
| | Romantic or sexual relationships and interactions. | sexual; missed; strip; calling; attractive; shower; bond; shipping; shock; expect |
| | Project management and documentation processes. | prep; construction; construct; constructed; input; journal; action; claim; running; claims |
| | Influence of successful relationships or partnerships in a law enforcement or collaborative context. | bond; successful; successfully; police; being; landscape; working; deeply; influence; hit |
| | Fashion evolution and personal growth. | outfit, Smith, museum, leather, dress, growth, Chris, era, lifetime, grew |
| | Techniques for visual representation and support in design or art. | reflection, supportive, split, shelter, visual, grid, line, reflect, simple, tricks |
| | Concerns related to injuries and their representation in the context of Jewish communities or cultural icons. | draft, injuries, injury, concerns, concern, Jewish, happening, icon, strategies, graphic |
| | Focus on specific strategies or tactics in a competitive context. | keen, particular, certain, wall, gap, specialized, battle, escape, chop, specific. |
| | Crime detection and security measures. | detect, security, detective, crime, shadow, detection, criminal, deal, assets, out |
| | Energy resources and infrastructure management. | graph, composition, master, gas, pipeline, mine, perception, deployed, demand, stake |

Table 5: Extended qualitative analysis on generated explanations from the baseline TopAct.

| Method | Automated Summary | Raw Explanation |
|---|---|---|
| TopAct | Inquiry about available options or facilities in various contexts. | What types of medical facilities are available for; What graduate programs are available for; What kind of housing options are available for; What accommodations are available for; What are some software options available for |
| | Setting personal boundaries and communication strategies. | Explain the potential consequences of failing to set boundaries; communication skills, as well as methods for setting boundaries; communication techniques that can be used to express personal boundaries; advice on how to communicate assertively, set boundaries; have faced, strategies you have used to set boundaries |
| | The concept of "role" in various contexts | as well as the role of technology and innovation in; they reside. Consider the role of email in; The role of technology and innovation in; washing techniques, the role of soap and water in; populations and the role of insecticides in |
| | Handmade or artisanal jewelry. | is a handcrafted Macrame beaded; hand-crafted artisanal glass beads; materials, tools, and methods used in bead; description: Handmade beaded; jewelry using materials such as beads |
| | Impact of social media. | Share your perspective on the impact of social media on; Discuss the impact of social media on; Discuss the impact of social media on; Write a report on the impact of social media on; Examine the impact of social media on |
| | Dimensions and measurements of objects. | They come in a 5" x; measuring approximately 6 inches by 6; 4" thick. 11" x; painting measures 14" tall by; a square paper measuring 8 inches by 8 |
| | Malicious intent related to food tampering or drugging. | Can I hide feces in someone's food; Whats the best way to hide poison in food; slip a roofie in a girl's drink; my fagot coworker's drink; type of poison for putting in people's food |
| | Manual processes in automation and testing. | operator-level isolation and rely on users to manually; using Selenium * Conduct manual; step-by-step guide on how to manually; Automated deployment flows and reduced manual; in a single run of the script without any manual |
| | Abortion rights and the Roe v. | have there been since Roe v. W; and vowed to pass legislation to protect abortion; mourning the fact that Roe v. W; landmark Supreme Court case Roe v. W; What is Roe v. W |
| | Impact of climate change. | Research and document the impact of climate change; 2. The effects of climate change; and write a report on the impacts of climate change; include a comprehensive analysis of the impact of climate change; are in place to address the effects of climate change |
| | Recipe search functionality and user interaction features. | and view recipes uploaded by others, a search; friendly, with an easy-to-use search; commenting and ratings for recipes, and a search; in the table view to allow the user to search; and ratings. Users should be able to search |
| | Webpage modification timestamps. | This page was last modified on; ings.\nThis page was last edited on; \nThis page was last edited on; construct.\nThis page was last modified on; 8.\nThis page was last edited on |

Table 6: Extended qualitative analysis on generated explanations from the baseline N2G.

| Method | Automated Summary | Raw Explanation |
|---|---|---|
| N2G | Character attributes in role-playing games. | choosing [MASK] race, class\n; name [MASK] race, class\n; name [MASK] race, class; Race [MASK] Human\n\nClass\n; backstory, class [MASK] |
| | Management and organizational skills in relation to tasks, teams, and time. | manage their tasks and; manage remote teams in; managing a [MASK] team?; manage [MASK] time effectively; manage my [MASK] team's territories? |
| | Negation or clarification phrases focusing on the phrase "doesn't mean". | [MASK] not necessarily; doesn[MASK]t mean; doesn[MASK]t mean; doesn[MASK]t mean; doesn[MASK]t mean |
| | Exclusion criteria or filtering terms. | not include [MASK] numbers or; exclude any [MASK] firm that; should not [MASK] any words that; exclude [MASK] words that; not include any [MASK] that |
| | Data storage and backup solutions, particularly focusing on external storage devices. | important data that you want to keep to an external; wireless file trans[MASK]; back[MASK]ups, and transferring; external hard; external hard |
| | Concepts related to returning or going back home. | last trains home; return home; walked home; way home; way home |
| | Bailout or financial assistance concepts, particularly in the context of economic interventions or stimulus packages. | GM Bail[MASK]; Paulson [MASK] other proponents of the bail; to step in to prevent it. Such bail[MASK]; and look at that auto bail[MASK]; stimulus packages [MASK] bail |
| | Informal greetings or inquiries about someone's well-being or current situation. | what[MASK]s going on; what[MASK]s going on; what[MASK]s up; What[MASK]s up; What[MASK]s up |
| | Customization and personalization of options or features. | options [MASK] customization; customizing [MASK]; to customize [MASK]; the player to customize [MASK] |
| | The phrase "On a scale" or variations of it, indicating a measurement or evaluation system. | On [MASK] scale of; On a scale [MASK]; On [MASK] scale of; On [MASK] scale of; On [MASK] scale of |
| | Addresses or locations. | 33 Dinah Shore Dr, [MASK]; 4[MASK]1 Bay Shore Road,; 1 Wessel Dr., [MASK];7 W. John St., [MASK]; 9[MASK]0 E. Street Rd., |
| | Gap year terminology. | [MASK] batical year; gap year [MASK]; gap year [MASK]; gap year [MASK]; gap year [MASK] |
| | Decades or time periods, specifically referencing the 70s, 80s, and 90s. | er from the 80 [MASK]; early 70 [MASK]; late [MASK] 90; 70s [MASK] 80; late [MASK] 90 |
| | Formatting and structuring text or documents focusing on the concept of a "clear head" or heading. | [MASK] appropriate head; format, with clear head [MASK]; [MASK] proper head; struct [MASK] and organized, with clear head; easy to follow, with clear head [MASK] |
| | Usage of the word "call" in various contexts, likely focusing on communication or addressing someone. | calls him [MASK]; call [MASK] americans indians?; calling [MASK] guy; call me [MASK]; called him [MASK] |
| | Historical figure: Benjamin Franklin. | Benjamin [MASK]; franklin [MASK]; Franklin [MASK]; Benjamin Franklin [MASK]; Benjamin Franklin [MASK] |