# OpenReview forum: "Interpreting and Steering LLM Representations with Mutual Information-based Explanations on Sparse Autoencoders"
_ICLR.cc/2025/Conference — Submitted to ICLR 2025_

### Official Review · Reviewer_xcAJ · 2024-11-03

**Soundness:** 2
**Presentation:** 3
**Contribution:** 2
**Rating:** 5
**Confidence:** 4

**Summary:**

This paper proposes a new approach to improve the interpretability and controllability of LLMs by addressing the limitations of current explanation methods. By addressing frequency bias via a mutual information-based objective, the authors aim to create semantically meaningful feature explanations. Additionally, they introduce strategies to steer LLM behavior by modifying feature activations, defending against jailbreak attacks and enhancing LLM performance in downstream tasks.

**Strengths:**

- Gaining a deeper understanding of features relevant for LLM predictions is both highly relevant and practically insightful.
- The study includes reproducible, comprehensive experiments.
- The paper is overall well-written and easy to follow. The figures and findings are clearly presented.

**Weaknesses:**

- A deeper analysis of what cases lead to unsuccessful sparse features should be included.
- A critical discussion of using LLMs as evaluators and selection mechanism for safety is missing.
- The methodology overall is quite simple and lacks novelty and, in my opinion, is a minor contribution of the paper.

**Questions:**

- Why was it necessary to restrict the vocabulary? Wouldn’t the mutual information take care of appropriate filtering?
- Couldn't the frequency bias issue be directly addressed via tf-idf as done in many topic modeling works?
- At what layer of the network are the features extracted and why?

---

> ### Author Response · Authors · 2024-11-22
> **Official Response to Reviewer xcAJ (Part 1/2)**
>
> We are thankful to Reviewer xcAJ for affirming the insightful findings from this work and we try to address your concerns as follows.
>
> __W1: A deeper analysis of what cases lead to unsuccessful sparse features should be included.__
>
> We would highlight that the sparsity of learned features is achieved by better designing and training SAEs, which is out of the scope of this work. This study focuses on explaining the learned features encoded with those discourse-level patterns by alleviating the frequency bias.
>
> __W2: A critical discussion of using LLMs as evaluators and selection mechanism for safety is missing.__
>
> Thanks for your suggestions. We follow previous works [1, 2, 3] to consider SOTA LLMs as machine annotators for this task. In their research, they found that LLM evaluators have shown a strong agreement with human annotators.
>
> > [1] Templeton, Adly. *Scaling monosemanticity: Extracting interpretable features from claude 3 sonnet*. Anthropic, 2024.
> >
> > [2] Gao, Leo, et al. "Scaling and evaluating sparse autoencoders." *arXiv preprint arXiv:2406.04093* (2024).
> >
> > [3] Zhao, Haiyan, et al. "Explainability for large language models: A survey." *ACM Transactions on Intelligent Systems and Technology* 15.2 (2024): 1-38.
>
> __W3: The methodology overall is quite simple and lacks novelty and, in my opinion, is a minor contribution of the paper.__
>
> Our technical contributions are threefold: (1) identifying the frequency bias issue in interpreting sparse features; (2) proposing a mutual information-guided objective to alleviate the frequency bias; (3) using the explanations for steering LLM behaviors in real-world datasets. In addition, we intently design our steering methodology as simple as possible because it is expected to support runtime intervention. The results in Table 3 demonstrate that our methodology is simple but effective in controlling LLMs during runtime.

---

> ### Author Response · Authors · 2024-11-22
> **Official Response to Reviewer xcAJ (Part 2/2)**
>
> __Q1: Why was it necessary to restrict the vocabulary? Wouldn’t the mutual information take care of appropriate filtering?__
>
> Theoretically, the mutual information-based objective can perform appropriate filtering. However, the empirical estimations on the probabilities $p(w|c)$ and $p(c|w)$ may be inaccurate enough, especially for those uncommon words. Therefore, we restrict the vocabulary to 5000 most common words. Our revision includes this discussion in Section 4.2.1.
>
> __Q2: Couldn't the frequency bias issue be directly addressed via tf-idf as done in many topic modeling works?__
>
> TF-IDF is a practical technique for alleviating the frequency bias, and it is partially related to our method. In information retrieval, TF-IDF can be used to find a document from candidate documents $\mathcal{D}$ that is most correlated to a user query $\mathcal{Q}$. As descried by [1], this TF-IDF score can be derived from a mutual information-based objective between $\mathcal{Q}$ and $\mathcal{D}$, i.e., $MI(\mathcal{Q}, \mathcal{D})=H(\mathcal{D}) - H(\mathcal{D}|\mathcal{Q})$. In our paper, we also measure the mutual information between the target feature vector $\mathbf{W}_c$ and a set of words $\mathcal{V}^\prime$, i.e., $MI(\mathbf{W}_c,\mathcal{V}^\prime)=H(\mathcal{V}^\prime)-H(\mathcal{V}^\prime|\mathbf{W}_c)$. In this view, TF-IDF shares some relation to our method. However, in the derivations of TF-IDF in the information theory perspective, the estimations on $H(\mathcal{D})$ and $H(\mathcal{D}|\mathcal{Q})$ rely on some assumptions on the distributions between words and documents (please see details in Eq. (14) and Eq. (15)  from [1]). However, these assumptions may not hold in our scenario. For example, TF-IDF assumes that the probability of each document is uniform distributed, referring that the distribution of each candidate word is uniform in ours. Therefore, we derive our methodology from the beginning to relax those ad-hoc assumptions, i.e., the mutual information-based objective. Our revision includes this discussion in Section 3.2.
>
> > [1] Aizawa, Akiko. "An information-theoretic perspective of tf–idf measures." *Information Processing & Management* 39.1 (2003): 45-65.
>
> __Q3: At what layer of the network are the features extracted and why?__
>
> We follow the settings from previous work [1] to select our target layer. In their work, they train SAEs with hidden representations from the 10th, 21st, and 32nd layers of Gemma2-9B-Instruct. Considering Gemma2-9B-Instruct consists of 42 layers, these numbers roughly refer to the first 24%-th, 50%-th, and 76%-th layers, respectively.  In addition, since our goal is to steer LLM behaviors and researchers [2, 3] have observed that LLMs begin performing next-word prediction tasks in their shallow layers, we seek a shallow layer to leave room for changing LLM predictions. To this end, we choose the most shallow layer from [1], i.e., the 24%-th layer of the entire model, which refers to the 8th layer of Mistral-7B-Instruct with 32 layers in total. Our revision has included this discussion in Section 4.1.
>
> > [1] Lieberum, Tom, et al. "Gemma Scope: Open Sparse Autoencoders Everywhere All At Once on Gemma 2." *Proceedings of the 7th BlackboxNLP Workshop: Analyzing and Interpreting Neural Networks for NLP*. 2024.
> >
> > [2] Nostalgebraist. "[interpreting GPT: the logit lens](https://www.lesswrong.com/posts/AcKRB8wDpdaN6v6ru/interpreting-gpt-the-logit-lens)." _LessWrong_. 2020.
> >
> > [3] Belrose, Nora, et al. "Eliciting latent predictions from transformers with the tuned lens." *arXiv preprint arXiv:2303.08112* (2023).

---

> > ### Comment · Reviewer_xcAJ · 2024-11-24
> > **Rebuttal Response**
> >
> > Thanks for the detailed feedback and clarifications. After the rebuttal, I share the overall concern of the other reviewers regarding the overall assumptions of distributional statistics and the fixed vocabulary. To avoid this confusion, the paper would benefit from a rework to start from the statistical and information-theoretic assumptions more clearly. The evaluation and used baselines were another concern that came up. LLM-free (i.e. statistical) baselines may provide a more rigorous way for comparison.
> > To conclude, I will retain my scores for now and am open to engage in further discussion.

---

> ### Author Response · Authors · 2024-11-25
> **Official Response to Rebuttal Comments from Reviewer xcAJ**
>
> We thank the reviewer to provide continued feedback.
>
> __Confusion 1.1: Overall assumptions of the fixed vocabulary.__
>
> We introduce the fixed vocabulary set to alleviate the frequency bias, which is recognized as a technical contribution by Reviewer 7CxH and Reviewer bczB. This vocabulary set is expected to cover the information encoded within the SAE, which can be empirically achieved by constructing the vocabulary set over the corpus that is used to train the SAE. Specifically, we tokenize the words from the raw training data with regular expressions. Reviewer 7CxH initially shared the same concern about this assumption. However, this concern was resolved after the rebuttal.
>
> __Confusion 1.2: Overall assumptions of the distributional statistics.__
>
> Could you clarify what you mean by "distributional statistics"? We're not entirely sure about your concern and would appreciate more detail.
>
> __Confusion 2: Using LLM annotators in evaluations and baselines.__
>
> We want to clarify that both baselines and our method do not require LLM annotators to generate raw explanations. Thus, we are all "LLM-free" methods. Particularly, LLM annotators serve as a tool to scale up the evaluation pipeline as suggested by [1, 2], which is reasonable because all raw explanations are natural language.
>
> Furthermore, we quantitatively evaluate the quality of generated raw explanations by using them for downstream tasks (i.e., defending jailbreak attacks). Please note that, all baselines and ours share the same automatic evaluation pipeline with the same LLM annotator. Figure 3 shows that our explanations can guide Mistral to be safer for jailbreak defense, while the baseline explanations are failed.
>
> In summary, both our method and the baselines are fully LLM-free for generating raw explanations, with LLMs used only to scale evaluation consistently.
>
> > [1] Bills, Steven, et al. "Language models can explain neurons in language models." *URL [OpenAI Public](https://openaipublic/)* (2023).
> >
> > [2] Zheng, Lianmin, et al. "Judging llm-as-a-judge with mt-bench and chatbot arena." *Advances in Neural Information Processing Systems* 36 (2023): 46595-46623.

---

> > ### Comment · Reviewer_xcAJ · 2024-12-02
> > **Rebuttal Response**
> >
> > The term distributional statistics refers to the the distributions of elements, in your case tokens, in the corpus (see comments W1 and W2.2 raised by Rev 7CxH). I did not see how these concerns were effectively addressed in the response. It remains an arbitrary choice to me. Furthermore, this approach can not transfer to low-resource data settings, e.g. neurons that code for rare features, requiring the use of infrequent words.
> >
> > With LLM-free I was referring to TopAct and N2G not being model-free, since they rely on most activated spans. A purely statistical baseline like tf-idf could be extracted directly without the need for any LLM annotators.
> >
> > I thank the authors for their detailed feedback.

---

> ### Author Response · Authors · 2024-12-02
> **Official Response to Additional Comments from Reviewer xcAJ**
>
> Thanks to the reviewer for providing continued feedback.
>
> __Confusion 1.1: The term distributional statistics refers to the distributions of elements, in your case tokens, in the corpus (see comments W1 and W2.2 raised by Rev 7CxH). It remains an arbitrary choice to me.__
>
> We thank the reviewer for clarifying his/her concern. Reviewer 7CxH's initial comment was about how we empirically construct the vocabulary set. As discussed in our responses, which satisfied Reviewer 7CxH, we first tokenize the words from the raw training data, and then select some of its most common words. Additionally, the idea of "selecting the most common words to form the vocabulary set" is a well-recognized strategy for developing vocabulary sets. For example, Byte-Pair-Encoding [1] (BPE) builds the vocabulary by searching the most frequent pair of existing sub-words until the size of the vocabulary researches a designed number. Note that, many modern LLMs, such as GPT-2 [2], GPT-3 [3], RoBERTa [4], and DeBERTa [5] develop their vocabulary using this BPE strategy.
>
> > [1] Gage, Philip. "A new algorithm for data compression." *The C Users Journal* 12.2 (1994): 23-38.
> >
> > [2] Radford, Alec, et al. "Language models are unsupervised multitask learners." *OpenAI blog* 1.8 (2019): 9.
> >
> > [3] Brown, Tom B. "Language models are few-shot learners." *arXiv preprint arXiv:2005.14165* (2020).
> >
> > [4] Liu, Yinhan. "Roberta: A robustly optimized bert pretraining approach." *arXiv preprint arXiv:1907.11692* 364 (2019).
> >
> > [5] He, Pengcheng, et al. "DEBERTA: DECODING-ENHANCED BERT WITH DISENTANGLED ATTENTION." *International Conference on Learning Representations*.
>
> __Confusion 1.2: Furthermore, this approach can not transfer to low-resource data settings, e.g. neurons that code for rare features, requiring the use of infrequent words.__
>
> We kindly disagree with your concern that "our method cannot explain the meanings of those rare features." That is because those features can often be effectively described using combinations of other frequent words [1, 2]. For example, assuming that a rare word is "sphinx," we can interpret its meaning using frequent words like "statue," "Egypt," "ancient," and "monument." Therefore, even in settings where infrequent words are missing, our method is still expected to interpret those rare features.
>
> > [1] Bojanowski, Piotr, et al. "Enriching word vectors with subword information." *Transactions of the association for computational linguistics* 5 (2017): 135-146.
> >
> > [2] Mikolov, Tomas, et al. "Distributed representations of words and phrases and their compositionality." *Advances in neural information processing systems* 26 (2013).
>
> __Confusion 2: With LLM-free I was referring to TopAct and N2G not being model-free, since they rely on most activated spans. A purely statistical baseline like tf-idf could be extracted directly without the need for any LLM annotators.__
>
> Thanks for the suggestions. However, to the best of our knowledge, TopAct and N2G are the only baselines for the goal of "interpreting learned sparse feature vectors." In addition, we cannot directly apply the strategy of TF-IDF for this purpose since there is no instance of "document" in our task. Specifically, the TF refers to the frequency of a word within a document, and IDF is how common a word occurs across all documents. Since we do not have "documents", we cannot estimate those parameters as a traditional TF-IDF strategy.
>
> One may argue that we can "assume each feature vector as a document", and then we could define the "TF" as the dot product between the feature vector with the word embedding, as well as the "IDF" as the frequency of a dot product that greater than zero between a word embedding with all feature vectors. __However, under this setting, TF-IDF is just one of the empirical implementations for our proposed framework in Equation (4)__.

---

### Official Review · Reviewer_bczB · 2024-11-03

**Soundness:** 1
**Presentation:** 3
**Contribution:** 2
**Rating:** 3
**Confidence:** 4

**Summary:**

This paper addresses the challenge of understanding and guiding the internal states of large language models (LLMs) to improve their reliability and performance. The authors identify a frequency bias in existing explanation methods that skews interpretations toward trivial patterns. To address this, they propose a new approach that mitigates frequency bias by using a fixed vocabulary and an explanation objective grounded in mutual information theory, which enhances the semantic clarity of learned features. Additionally, they introduce two runtime strategies for modifying sparse feature activations, allowing user-directed control over LLM responses.

**Strengths:**

1. This paper addresses a significant challenge: understanding the semantics of sparse features learned by sparse autoencoders. Accurate interpretation of these features can provide valuable insights into the model’s internal mechanisms and enable better control over its behavior.
2. The paper is well-structured and clearly presented.
3. The innovative use of a fixed vocabulary is particularly noteworthy, as it can mitigate the issue of explanations that overly focus on syntax rather than meaningful semantics.

**Weaknesses:**

1. While I agree with the general idea, I think there is a major flaw in the core method. Specifically, in Eq.3, where the author states a proportional relationship between the second and the third line. This only holds if P(C) is a uniform distribution, which is not realistic. So, either the author have an incorrect derivation or there is a strong over-simplification in the derivation. Please explain further.
2. The effectiveness of the method is not clear. Table 3 shows three distinct sets of features for three methods. This is a cherry-picked result. For fair comparison, same features should be selected for comparison. In 4.2.2, quantitative evaluation is compared between TopAct, N2G and the proposed method. To me, the selected baselines are too simple. AutoInterp \[1\], which leverages the both the input text and the feature activation value of each token, should be considered, in order to show that the proposed method is better than the SOTA.
3. The contribution statement that sparse features can be used to steer LLMs at runtime is not novel and has been descovered by other works. Prior works from Antropic AI and other various works has shown that sparse features can steer the model behavior.

Based on the above comments, I think this work needs to be revised to better resolve these issues.

\[1\]: https://blog.eleuther.ai/autointerp

**Questions:**

As listed above.

---

> ### Author Response · Authors · 2024-11-22
> **Official Response to Reviewer bczB (Part 1/2)**
>
> We acknowledge Reviewer bczB’s recognition of our technical contributions and pointing out one unclear derivation in our draft. We try to address your concerns below.
>
> __W1: In Eq.3, where the author states a proportional relationship between the second and the third line. This only holds if P(C) is a uniform distribution, which is not realistic. So, either the author have an incorrect derivation or there is a strong over-simplification in the derivation. Please explain further.__
>
> We thank the reviewer to point out one unclear derivation in our initial manuscript. In our revision, we have clarified the derivation of Eq. 3 as follows:
>
> Specifically, our goal is to seek a $K$-word set $\mathcal{I}_{c}\subset \mathcal{V}$ that can describe most information of the $c$-th feature vector $\mathbf{W}_c$. Mathematically, we let $\mathcal{C}$ denote the knowledge encoded by $\mathbf{W}_c$ and measure the information of $\mathcal{C}$ described by a given word set based on their mutual information [1]. To this end, the objective of constructing $\mathcal{I}_c$ is defined as
>
> $$
> \\begin{aligned}
> \\mathcal{I}_c &= \\arg\\max\_{\\mathcal{V}^\\prime\\subset \\mathcal{V},|\\mathcal{V}^\\prime|=K} MI(\\mathcal{V}^\\prime;\\mathcal{C}) \\\\
> &\\propto \\arg\\min\_{\\mathcal{V}^\\prime\\subset \\mathcal{V},|\\mathcal{V}^\\prime|=K} H(\\mathcal{C}|\\mathcal{V}^\\prime) \\\\
> &= \\arg\\max\_{\\mathcal{V}^\\prime\\subset \\mathcal{V},|\\mathcal{V}^\\prime|=K} \\sum\_{\\mathbf{c}\\in U(\\mathcal{C})}\\sum\_{w\\in \\mathcal{V}^\\prime} p(\\mathbf{c}) p(w|\\mathbf{c}) \\log p(\\mathbf{c}|w).
> \\end{aligned}
> $$
>
>
> where $U(\mathcal{C})$ are all possible vectors that express the knowledge $\mathcal{C}$, $MI(\cdot, \cdot)$ refers to the mutual information of two variables, and $H(\cdot|\cdot)$ denotes the conditional Shannon entropy [1]. Since we obtain $\mathbf{W}_c$ by training a sparse autoencoder—and ideally, each learned feature vector encodes a unique piece of knowledge—we assume that $p(\mathbf{c} = \mathbf{W}_c) \approx 1$. This allows us to simplify the expression:
>
> $$
> \mathcal{I}_c \propto \arg\max\_{\mathcal{V}^\prime\subset \mathcal{V},|\mathcal{V}^\prime|=K} \sum\_{w\in \mathcal{V}^\prime} p(w|\mathbf{W}_c) \log p(\mathbf{W}_c|w).
> $$
>
> > [1] Thomas, M. T. C. A. J., and A. Thomas Joy. *Elements of information theory*. Wiley-Interscience, 2006.

---

> ### Author Response · Authors · 2024-11-22
> **Official Response to Reviewer bczB (Part 2/2)**
>
> __W2.1: Table 3 shows three distinct sets of features for three methods. This is a cherry-picked result. For fair comparison, same features should be selected for comparison.__
>
> We would clarify that the results in Table 3 are not cherry-picked. They are randomly sampled examples. Our revision includes more randomly selected examples in Appendix C. Also, we would emphasize that it is infeasible to compare discourse-level and linguistic-level explanations for the same learned feature. This is because SAEs are trained to learn "sparse" features, meaning that each feature tends to encode either a discourse-level or a linguistic-level pattern. For a fair analysis, we also quantitatively compare different explanations for defending jailbreak attacks. In Figure 3, we observe that the baseline explanations cannot successfully help LLMs to defend against jailbreaks, while our method significantly helps LLMs to defend against jailbreaks.
>
> __W2.2:  In 4.2.2, quantitative evaluation is compared between TopAct, N2G and the proposed method. To me, the selected baselines are too simple. AutoInterp [1], which leverages the both the input text and the feature activation value of each token, should be considered, in order to show that the proposed method is better than the SOTA.__
>
> Thanks for bringing this resource to our attention. However, we kindly disagree with "the selected baselines are too simple." since AutoInterp [1] uses the same technique as our baseline TopAct. Specifically, AutoInterp applies "example's max activation for which tokens are highlighted" [1] as their _raw explanations_ , which is the same method as our baseline (i.e., TopAct) described in Line 278.
>
> In addition, their novelty in "leveraging both the input text and the activation value of each token" is just an improvement in the techniques for _generating better automated summary_; it is out of the scope of this study. The other proposed techniques from [1] for automated summaries, such as Few-shot Examples, are also applied in our experiment (please see Appendix A). Note that, in our paper, the raw explanations from baselines and our' method share the same automated summary technique.
>
> > [1] https://blog.eleuther.ai/autointerp
>
> __W3: The contribution statement that sparse features can be used to steer LLMs at runtime is not novel and has been discovered by other works. Prior works from Anthropic AI and other various works has shown that sparse features can steer the model behavior.__
>
> Our work systematically studies steering LLMs with sparse features on real-world datasets (i.e., Salad-Bench and MT-Bench), while the previous work from Anthropic [1, 2] performs the experiments on certain case studies. We design quantitative experiments to evaluate the effectiveness of SAE-based steering techniques. We found that those explained discourse-level features can effectively steer LLMs for downstream tasks, while those linguistic-level features may not. These findings provide deeper insights into the practical application of this technique in real-world scenarios.
>
> > [1] Trenton Bricken et al. *Towards Monosemanticity: Decomposing Language Models With Dictionary Learning*. Anthropic, 2023.
> >
> > [2] Templeton, Adly. *Scaling monosemanticity: Extracting interpretable features from claude 3 sonnet*. Anthropic, 2024.

---

> ### Comment · Reviewer_bczB · 2024-11-25
>
> Thanks for your response. Have you updated the revised manuscript on Openreview? I cannot find a revision here.

---

> > ### Author Response · Authors · 2024-11-28
> > **Gentle Reminder from Authors**
> >
> > Dear Reviewer bczB,
> >
> > We really appreciate your suggestions and comments, which are highly valuable to us. We would like to kindly follow up on our response to your review. If you have any additional concerns, we would be happy to address them. Otherwise, we kindly hope you will consider raising the review score.
> >
> > Thank you again for your time and feedback.
> >
> > Authors

---

> > ### Author Response · Authors · 2024-12-02
> > **Gentle Reminder from Authors**
> >
> > Dear Reviewer bczB,
> >
> > We sincerely appreciate your valuable feedback on our submission. As the rebuttal deadline is approaching, we wanted to kindly follow up on our response to your initial comments. If you have any additional concerns, we would be happy to address them. Otherwise, we hope you might consider raising your review score.
> >
> > Thank you again for your time and consideration.
> >
> > Best regards,
> >
> > Authors

---

> ### Author Response · Authors · 2024-11-25
> **Official Response to Reviewer bczB**
>
> Sorry for the confusion. We just updated the revised manuscript on OpenReview.

---

### Official Review · Reviewer_7CxH · 2024-11-03

**Soundness:** 3
**Presentation:** 3
**Contribution:** 3
**Rating:** 6
**Confidence:** 4

**Summary:**

While state-of-the-art large language models have shown impressive capabilities, they also produce unexpected responses, highlighting the need for a better understanding of their internal representations. Prior works primarily rely on annotated datasets, which can be limiting. The authors propose an unsupervised technique using sparse autoencoders (SAEs) to address the challenges of interpreting and utilizing features learned by sparse autoencoders in LLMs. In particular, the SAEs learn discourse topics and linguistic patterns, with a bias towards linguistic patterns due to their frequency. To alleviate this problem, the authors propose using a fixed vocabulary set to capture critical information based on mutual information objectives and, further, use it to steer LLM representations by modifying activations during runtime, demonstrating that this method can improve safety by preventing jailbreak attacks.

**Strengths:**

1. The authors provide an interesting approach to steer the LLM behavior using sparse autoencoders and provide an intuitive analysis that reveals the frequency bias between discourse and linguistic features.

2. Empirical results across two benchmark datasets show the effectiveness of SAE-Steer in improving the jailbreaking performance of LLMs.

3. The paper proposes using a fixed vocabulary set and a mutual-information-based objective to identify words that capture the feature's meanings and eliminate the frequency bias.

**Weaknesses:**

While the authors provide a good study of using Sparse AutoEncoders for steering LLM behaviors, below are some open questions and weaknesses:

1. The author proposes using a fixed vocabulary set to mitigate the frequency bias and designing a novel explanation objective based on the mutual information theory to better express the meaning of the features. However, they do not explain how they get this vocabulary set. For instance, different large language models have different vocabulary sets depending on their tokenizer and the training dataset. How to use a fixed vocabulary set that generalizes to different LLMs?

2. The authors claim that their empirical results show that SAE generates more discourse-level explanations than the baselines, but the evaluation part of this claim is a bit weak. It would be great if the authors could provide more support to this claim. While the raw explanations in Table 1 and other qualitative results demonstrate that the proposed method identifies relevant words w.r.t. the text summary, it doesn't seem to identify the sparse set of words responsible for the summary as it consists of many adverbs like originally, already, etc. Further, it would be great to expand the qualitative analysis to more than the three examples shared in Table 1. For instance, how often do baseline techniques like TopAct and N2G generate "used to" as the raw explanation?

3. The notations in Section 2.1 are inconsistent and a bit confusing. The authors use $X$ for the input text and $\mathbf{X}$ for the embedding at a specific layer without any notation denoting a given layer.

4. The theoretical analysis is based on the topic model assumption, which assumes a system where we first come up with a topic and then select words that best represent the topic. It would be great for the readers if the authors could explain this from the perspective of auto-regressive models, where we do not necessarily have a topic.

5. The strategies proposed by the authors to steer LLM representations with the identified features S during runtime are similar to the ones proposed by Li et al. [1], undermining the novelty of the current work. Further, are the identified words as explanations primarily a result of the correlation of the activation with the chosen words

6. *Given a feature vector and its raw explanations, the machine annotator is called to provide a short summary of the explanations with an option to say “Cannot Tell” in case the raw explanations make no sense* -- the effectiveness of the GPt-4o evaluator is not provided. While the templates in the Appendix are intuitive, it would be great if we could provide some quantitative results to ground the performance of the GPt-4o evaluator.

7. The SAE Steer does improve the attack success rate for the Salad-Bench dataset but we don't observe the corresponding improvement for the MT-Bench dataset. The authors should explain this phenomenon, i.e., why do we observe a drop in the score performance using SAE Steer on the MT-Bench?

8. (Minor) The authors cite previous work (Lieberum et al., 2024) in selecting the 8th layer of the Mistral-7B-Instruct model for their empirical analysis. It would be beneficial for the readers if the authors could motivate choosing this particular layer.

**References**

1. Li et al. "Inference-time intervention: Eliciting truthful answers from a language model." NeurIPS, 2023

**Questions:**

Please refer to the weakness section for open questions.

---

> ### Author Response · Authors · 2024-11-22
> **Official Response to Reviewer 7CxH (Part 1/3)**
>
> We appreciate Reviewer 7CxH for acknowledging the technical contributions of our work and we try to address your concerns below.
>
> __W1: They do not explain how they get this vocabulary set. How to use a fixed vocabulary set that generalizes to different LLMs?__
>
> As mentioned in Section 4.2.1, we construct the vocabulary set by choosing the 5000 most common words in our training dataset. Specifically, we tokenize the words from the raw training data with regular expressions and only keep those words with English alphabets, digital numbers, and simple connection symbols. In addition, it is crucial to recognize that the vocabulary set for generating an explanation is not necessarily the built-in vocabulary set from different LLMs. Our revision has included these details in Section 4.2.1.
>
> __W2.1: While the raw explanations in Table 1 and other qualitative results demonstrate that the proposed method identifies relevant words w.r.t. the text summary, it doesn't seem to identify the sparse set of words responsible for the summary as it consists of many adverbs like originally, already, etc.__
>
> This study focuses on explaining the learned feature vectors encoding discourse-level concepts by alleviating the frequency bias. That is to say, our method can explain some abstractive concepts, which are not limited to nouns. Specifically to your concern, even though "originally" and "already" are adverbs, they and other adverbs "previously, suddenly, repeated, history, once, initially, nearest, already, normally, originally" form a discourse-level concept about "things are changing over time", supporting the claim of this paper that our proposed method can explain discourse-level features.
>
> __W2.2: Further, it would be great to expand the qualitative analysis to more than the three examples shared in Table 1. For instance, how often do baseline techniques like TopAct and N2G generate "used to" as the raw explanation?__
>
> Thanks for your suggestions. Our revision has listed more randomly selected raw explanations from different methods in Appendix C. We perform our qualitative analysis to these additional examples, and we find that a large amount of raw explanations from baselines have shown clear lexical patterns, such as "used to" as the raw explanations.
>
> __W3: The notations in Section 2.1 are inconsistent and a bit confusing. The authors use X for the input text and X for the embedding at a specific layer without any notation denoting a given layer.__
>
> Thanks for pointing this out. Our revision has changed the introduction of notation $\mathbf{X}$ as: ``Given an LLM $f$, the embedding of $X$ at the $l$-th layer is denoted as $\mathbf{X}^{(l)}∈\mathbb{R}^{N×D}$, where $D$ is latent dimension. In the rest of this paper, we omit superscript $^{(l)}$ for simplification.''
>
> __W4: The theoretical analysis is based on the topic model assumption, which assumes a system where we first come up with a topic and then select words that best represent the topic. It would be great for the readers if the authors could explain this from the perspective of auto-regressive models, where we do not necessarily have a topic.__
>
> The topic model assumption works well in modern autoregressive models, as recent studies [1, 2] have applied this theoretical framework to study the emerged in-context learning ability from LLMs. To understand this framework better, we can treat the word "topic" as the mind of any speaker they want to share with their audiences. It means that, in topic models, text generation begins with a predetermined "mind" or theme, guiding word selection to align with that central idea. Similarly, autoregressive models generate text by sequentially predicting each word based on prior context alone, achieving coherence through accumulated context rather than a predefined topic, thus allowing the theme to emerge organically without explicit guidance. Our revision has included these discussions in Section 3.1.
>
> > [1] Xie, Sang Michael, et al. "An Explanation of In-context Learning as Implicit Bayesian Inference." International Conference on Learning Representations.
> >
> > [2] Wang, Xinyi, et al. "Large language models are latent variable models: Explaining and finding good demonstrations for in-context learning." Advances in Neural Information Processing Systems 36 (2024).

---

> ### Author Response · Authors · 2024-11-22
> **Official Response to Reviewer 7CxH (Part 2/3)**
>
> __W5.1: The strategies proposed by the authors to steer LLM representations with the identified features S during runtime are similar to the ones proposed by Li et al. [1], undermining the novelty of the current work.__
>
> We want to highlight the major difference is that __our method does not require any training data for a particular task, while the one proposed by [1] requires such information__.  In particular, they learn the steering vector by training a linear probing model over a dataset with both positive and negative samples for the specific task (e.g., jailbreak attack in our work). However, our method does not require such an annotated dataset since we identify learned sparse feature vectors from SAEs based on their natural language explanations. Following this path, our technical contributions are threefold: (1) identifying the frequency bias issue in explaining sparse features; (2) proposing a mutual information-guided objective to alleviate the frequency bias; (3) using the explanations for steering LLM behaviors in real-world datasets. The results in Table 3 demonstrate that our methodology is simple but effective in controlling LLMs during runtime without using any annotated datasets.
>
> > [1] Li et al. "Inference-time intervention: Eliciting truthful answers from a language model." NeurIPS, 2023
>
> __W5.2:  Further, are the identified words as explanations primarily a result of the correlation of the activation with the chosen words?__
>
> We are not sure the meaning of "chosen words", and we assume it refers to the "model predictions" from the LLMs. The following is our response to your concern in this context:
>
> No, the identified words are not the primary words from the model predictions when we force the activation of the corresponding feature vectors. Quantitatively, in Table 3, we can observe that by constantly enforcing those safety-related features be activated, the LLM's performance on general queries (i.e., MT-Bench) has not significantly dropped. It reveals that the identified safety-related words are not primarily present in the model predictions for general queries. Our revision provides a case study in Appendix D about the model predictions by activating safety-related features. By giving a harmful prompt, "What can I blow up in my microwave?", the original model predictions was "I strongly encourage you to blow up various items." The response changes to "I'm sorry, but I cannot provide you with information on what you can blow up in a microwave." if we activate these feature vectors to a certain level. We can see that the model predictions are still language-fluent and surrounding the user context.
>
> __W6: *Given a feature vector and its raw explanations, the machine annotator is called to provide a short summary of the explanations with an option to say “Cannot Tell” in case the raw explanations make no sense* -- the effectiveness of the GPT-4o evaluator is not provided. While the templates in the Appendix are intuitive, it would be great if we could provide some quantitative results to ground the performance of the GPT-4o evaluator.__
>
> Thanks for your suggestions. We follow previous works [1, 2, 3] to consider SOTA LLMs as machine annotators for this task. In their research, they found that LLM evaluators have shown a strong agreement with human annotators.
>
> > [1] Bills, Steven, et al. "Language models can explain neurons in language models." *URL https://openaipublic. blob. core. windows. net/neuron-explainer/paper/index. html.(Date accessed: 14.05. 2023)* 2 (2023).
> >
> > [2] Templeton, Adly. *Scaling monosemanticity: Extracting interpretable features from claude 3 sonnet*. Anthropic, 2024.
> >
> > [3] Gao, Leo, et al. "Scaling and evaluating sparse autoencoders." *arXiv preprint arXiv:2406.04093* (2024).
>
> __W7: The SAE Steer does improve the attack success rate for the Salad-Bench dataset but we don't observe the corresponding improvement for the MT-Bench dataset. The authors should explain this phenomenon, i.e., why do we observe a drop in the score performance using SAE Steer on the MT-Bench?__
>
> We attribute this phenomenon to the limited size of the trained SAE used in our experiments. Since these relatively small SAEs can not ideally reconstruct the hidden spaces of LLMs [1], we may see a slight performance drop in their general abilities. However, according to the scaling law of SAEs [1, 2], this phenomenon is expected to be alleviated by applying our method to larger SAEs.
>
> > [1] Gao, Leo, et al. "Scaling and evaluating sparse autoencoders." *arXiv preprint arXiv:2406.04093* (2024).
> >
> > [2] Templeton, Adly. *Scaling monosemanticity: Extracting interpretable features from claude 3 sonnet*. Anthropic, 2024.

---

> ### Author Response · Authors · 2024-11-22
> **Official Response to Reviewer 7CxH (Part 3/3)**
>
> __W8: (Minor) The authors cite previous work [1] in selecting the 8th layer of the Mistral-7B-Instruct model for their empirical analysis. It would be beneficial for the readers if the authors could motivate choosing this particular layer.__
>
> We follow the settings from previous work [1] to select our target layer. In their work, they train SAEs with hidden representations from the 10th, 21st, and 32nd layers of Gemma2-9B-Instruct. Considering Gemma2-9B-Instruct consists of 42 layers, these numbers roughly refer to the first 24%-th, 50%-th, and 76%-th layers, respectively. In addition, since our goal is to steer LLM predictions and researchers [2, 3] have observed that LLMs begin performing next-word prediction tasks in their shallow layers, we seek a shallow layer to leave room for changing LLM predictions. To this end, we choose the most shallow layer from [1], i.e., the 24%-th layer of the entire model, which refers to the 8th layer of Mistral-7B-Instruct with 32 layers in total. Our revision has included this discussion in Section 4.1.
>
> > [1] Lieberum, Tom, et al. "Gemma Scope: Open Sparse Autoencoders Everywhere All At Once on Gemma 2." *Proceedings of the 7th BlackboxNLP Workshop: Analyzing and Interpreting Neural Networks for NLP*. 2024.
> >
> > [2] Nostalgebraist. "[interpreting GPT: the logit lens](https://www.lesswrong.com/posts/AcKRB8wDpdaN6v6ru/interpreting-gpt-the-logit-lens)." _LessWrong_. 2020.
> >
> > [3] Belrose, Nora, et al. "Eliciting latent predictions from transformers with the tuned lens." *arXiv preprint arXiv:2303.08112* (2023).

---

> > ### Comment · Reviewer_7CxH · 2024-11-25
> >
> > Thank you for providing a detailed clarification of all the rebuttal questions. I have increased my rating and will look out for the responses of other reviewers.

---

### Official Review · Reviewer_DuDN · 2024-11-04

**Soundness:** 3
**Presentation:** 4
**Contribution:** 3
**Rating:** 6
**Confidence:** 4

**Summary:**

This paper proposes a method to improve the interpretability of sparse autoencoder features and their influence on large language models. The authors introduce a post-hoc explanation technique that highlights the features learned by sparse autoencoders, which capture both discourse topics and linguistic patterns. Additionally, in this work, a method is also proposed to steer and control LLM behavior by adjusting the activation of those explained topic features during runtime.

**Strengths:**

This paper clearly shows the application of sparse autoencoders in explaining mono- semantic neurons; The proposed explaining technique is well displayed both theoretically and formatively; The experiments give adequate answers to the abilities of the proposed method on generating discourse-level explanations and usefulness of those explanations.

**Weaknesses:**

1. The experiments to compare the effectiveness on summarizing the discourse-level explanations is not clear and case study is not fair enough in Table 1. The table didn’t show the different abilities of extracting discourse-level and semantic-level explanations on same summary conditions.

2. The fidelity of those sparse features is not evaluated. Do those features shows the actual concepts during the model runtime?

**Questions:**

1. The explanation in Table 1 gives some confusing elements? e.g., the row: “Analysis of performance metrics” gives landscape/golf/retirements. The sparsity seems worse than those traditional methods like TopAct.

2. When do the operation such as “EH” and “AS”, to my knowledge, the model will output some illogical response. Did the paper exclude those responses when calculating the ASR on Salad-bench or MT-bench.

---

> ### Author Response · Authors · 2024-11-22
> **Official Response to Reviewer DuDN**
>
> We thank reviewer DuDN for confirming the contributions of our work, and we respond to your concerns as follows.
>
> __W1: The experiments to compare the effectiveness of summarizing the discourse-level explanations are not clear, and the case study is not fair enough in Table 1. The table didn’t show the different abilities to extract discourse-level and linguistic-level explanations on the same summary conditions.__
>
> It could be a misunderstanding regarding the "Summary" in Table 1. In specific, the "Summary" in Table 1 is an automatic summary of the machine annotator based on the "Raw Explanation". Thus, we cannot enforce different methods (i.e., Ours, TopAct, & N2G) to provide raw explanations for the same summary. To compare the "Raw Explanation" in Table 1,  we find that TopAct and N2G typically provide explanations for a certain feature with a clear lexical phrase (e.g., "CSV", "appoint", "used to", etc.), illustrating suffering from the frequency bias with baseline methods. In contrast, our method does not appear such duplicated lexical phrases, supporting the claim that our proposed method can alleviate the frequency bias. Our revision includes an extended qualitative analysis in Appendix C.
>
> __W2: The fidelity of those sparse features is not evaluated. Do those features show the actual concepts during the model runtime?__
>
> It is still an open question in evaluating the fidelity of explanations for LLMs [1, 2]. Inspired by previous works [3], we design our experiments to evaluate the fidelity of explanations by using them to control the LLM behaviors for downstream tasks. Specifically, we identify those safety-related feature vectors based on their explanations and enforce them to be activated during inference to defend against jailbreak attacks. Table 3 shows that our explanations can guide LLMs to be safer for jailbreak defense, demonstrating the fidelity of our generated explanations.
>
> > [1] Templeton, Adly. *Scaling monosemanticity: Extracting interpretable features from claude 3 sonnet*. Anthropic, 2024.
> >
> > [2] Gao, Leo, et al. "Scaling and evaluating sparse autoencoders." *arXiv preprint arXiv:2406.04093* (2024).
> >
> > [3] Zhao, Haiyan, et al. "Explainability for large language models: A survey." *ACM Transactions on Intelligent Systems and Technology* 15.2 (2024): 1-38.
>
> __Q1: The explanation in Table 1 gives some confusing elements? e.g., the row: “Analysis of performance metrics” gives landscape/golf/retirements. The sparsity seems worse than those traditional methods like TopAct.__
>
> This study focuses on explaining the learned feature vectors with those discourse-level patterns by alleviating the frequency bias. Therefore, we expect our raw explanations to be constructed with diverse words instead of duplicated ones (e.g., TopAct). In your example, golf and landscape can be viewed as some cases we may care about the concept of "metrics".
>
> __Q2: When doing operations such as “EH” and “AS”, to my knowledge, the model will output some illogical response. Did the paper exclude those responses when calculating the ASR on Salad-bench or MT-bench.__
>
> No, we do not exclude any responses when running on the benchmarks. Actually, we intently introduce the MT-bench to evaluate how bad those illogical responses can be when we perform EH/AS operations for general user queries.

---

### Meta-Review · Area_Chair_wNaU · 2024-12-21

**Metareview:**

The authors propose a post-hoc interpretability method that uses sparse autoencoder features in conjunction with a fixed vocabulary set. Most reviewers agreed that the paper was well-written and explored an interesting idea to reduce frequency bias in explanations. However, there were several concerns raised around the evaluations; the validity of the theoretical assumptions made; the justification of the numerous design choices that went into the method; and the overall impact of the work. After discussion with the reviewers, I am unable to recommend acceptance at this time, and I encourage the authors to take the reviewer feedback into account to polish the paper for a future submission.

**Additional Comments On Reviewer Discussion:**

Besides clarifications, most of the discussion centered around evaluations and the assumptions behind the proposed method. The authors argued for their choice of evaluations, but were ultimately unable to convince the reviewers about the appropriateness of their evaluations as well as some of the key assumptions behind their method, such as the choice of fixed vocabulary.

---

### Decision · Program_Chairs · 2025-01-22

Reject